# Mitigation of Calibration Ringing in the Context of the MTG-S IRS Instrument

**Pierre Dussarrat [1,*], Guillaume Deschamps [1], Bertrand Theodore [1], Dorothee Coppens [1], Carsten Standfuss [2] and Bernard Tournier [2]**

1   EUMETSAT, Eumetsat-Allee 1, 64295 Darmstadt, Germany
2   SPASCIA, 14 Avenue de l'Europe, 31520 Ramonville-Saint-Agne, France
*   Correspondence: pierre.dussarrat@eumetsat.int

**Abstract:** EUMETSAT is currently developing the on-ground processing chain of the infrared sounders (IRS) on-board the Meteosat third-generation sounding satellites (MTG-S). In this context, the authors investigated the impact of a particular type of radiometric error, called hereafter calibration ringing. It arises in Fourier transform spectrometers when the instrument's radiometric transfer function (RTF) varies within the domain of the instrument's spectral response function (SRF). The expected radiometric errors were simulated in the context of the MTG-S IRS instrument in the long-wave infrared (LWIR) band. Making use of a principal components (PCs) decomposition, a software correction, called RTF uniformisation, was designed and its performance was assessed in the context of MTG-S IRS.

**Keywords:** Meteosat third generation; IRS; Fourier transform spectrometry; radiometric calibration; ringing; principal components

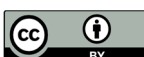

## 1. Introduction

Fourier transform infrared spectrometry from space allows for decomposing the light exiting the atmosphere from its top: the reconstructed spectra exhibit absorption and emission lines representative of the Earth's atmospheric composition and thermodynamic state. Instruments, such as the infrared atmospheric sounding interferometer (IASI) on Metop [1] and the cross-track infrared sounder (CrIS) on Suomi-NPP [2], have proven to be very effective for both weather forecasting and climate monitoring. EUMETSAT is now preparing for the next generation of instruments and, in particular, for the first European Fourier transform spectrometer in a geostationary orbit, the infrared sounder (IRS) on-board the Meteosat third-generation sounding satellites (MTG-S) [3]. The ambition to generate spectra with a radiometric accuracy below the deci-Kelvin limit has brought attention to a subtle radiometric bias in Fourier transform spectrometry that we refer to hereafter as calibration ringing [4,5]. Such an effect arises when an instrument's radiometric transfer function (RTF), sometimes also referred to as the spectral instrument responsivity, varies significantly within the domain of the instrument's spectral response function (SRF). These variations generate distortions in the SRF [4], which, if unaccounted for, propagate as radiometric errors into the calibrated Earth-view spectra exploited by the users.

RTF spectral variations are actually expected for most instruments. As a first example, transmission cut-offs of optical elements could lead to strong transmission gradients at the band edges and would produce calibration ringing. Thus, Borg et al. [6] recently reported on the presence of such ringing in spectra measured with the CrIS instrument that are induced by a steep instrument RTF at the beginning of the LWIR band. As a second example, RTF modulations can arise from unexpected light loops between optical

surfaces. Optical elements in transmission, such as lenses, windows or protective layers with non-perfect coating, can create low-finesse etalons [7]; as a result, the transmission appears to be modulated as function of the incident light wavenumber. The latter effect is expected in the MTG-S IRS instrument with a spectral modulation with a periodicity of approximately 250 m⁻¹ (modulation frequency of 4 mm) and up to a few percent relative amplitude. The optimization of the optical design of the instrument (e.g., redesigning the optical coating such that the RTF is flat) has proven to be technically out of reach. Therefore, assuming a frozen design of the IRS instrument, the authors initiated a study aiming at developing a methodology to mitigate the impact of calibration ringing and to introduce the correction into the operational on-ground processing.

The fundamentals of calibration ringing are recalled in Section 2. The discussion and definition of the mitigation strategy, called RTF uniformisation, follows in Section 3. Finally, the last section presents simulations testing the RTF uniformisation in the context of the MTG-S IRS long-wave infrared band (LWIR).

## 2. Calibration Ringing

Calibration ringing errors occur when radiometric calibration fails to perfectly compensate for RTF spectral variations. Usually, an instrument's optical transmission is characterized in flight with specific calibration schemes using, for example, on-board blackbody and deep-space measurements [8]. The optical transmission is then removed from the raw Earth-view measurements by division by the radiometric calibration slope. However, we still expect the occurrence of calibration ringing as high-frequency residual spectral modulations. In the following, we propose a simplified description of the radiometric calibration step of FTS products to highlight the genesis of calibration ringing.

The instrument SRF, noted as $SRF(v)$ as function of the wavenumber, $v$, is given in the context of Fourier transform spectrometry by the Fourier transform of the numerical apodisation, $Apod(x)$, applied to the recorded interferograms as function of the optical path distance, $x$ (OPD), multiplied by the potential wavenumber-dependent self-apodisations induced by instrumental defects. We assume hereafter, for simplicity, that the self-apodisation defects are negligible or perfectly compensated by dedicated software corrections such that the SRF is the same for all spectral channels [9,10].

The radiometric calibration slope coefficient is usually computed from the ratio between a spectrum measured from a black-body view and the associated theoretical Planck radiance at the blackbody temperature, $\Gamma_{TBB}(v)$. As the blackbody radiation is flat at the scale of the SRF, the radiometric calibration slope coefficient $R(v)$ can be written as the RTF convoluted with the SRF only:

$$R(v) = \frac{[\Gamma_{TBB}.T \otimes SRF](v)}{\Gamma_{TBB}(v)} \cong [T \otimes SRF](v) \tag{1}$$

where $T(v)$ is the RTF of the instrument. If $Sp(v)$ is the radiance spectrum exiting the atmosphere as a function of the wavenumber then the raw spectra are given by $[Sp.T \otimes SRF](v)$, and the radiometrically calibrated radiances $Sp_r(v)$ are written as:

$$Sp_r(v) \cong \frac{[Sp.T \otimes SRF](v)}{[T \otimes SRF](v)} \neq [Sp \otimes SRF](v) \tag{2}$$

Thus, if the optical transmission varies significantly at the scale of the SRF then the calibrated spectrum does not equal the input spectrum convoluted with a unique, pixel-independent SRF, as desired by the users of FTS-based multi-detector data products such as those of IASI, CrIS, and IRS. The difference between the two terms defines the so-called calibration ringing error [4]. As illustrated in this paper, calibration ringing radiometric errors usually appear as both a modulation and spikes that are functions of the input scene. Therefore, it cannot be canceled completely by a simple bias correction of the calibrated spectra and may affect the retrieval of the atmospheric composition.

As discussed in [6], the effect of calibration ringing in data applications does not necessarily constitute an error if the equivalent SRF distortions are explicitly accounted for in radiative transfer models (RTM), cf. Equation (2). This is even without an alternative for grating spectrometers (such as AIRS), for which the generation of data to a common spectral response across detectors is unfeasible. However, if the transmission were to vary between detectors, the SRF would become detector-dependent. In the case of the CrIS instrument, Borg et al. [6] demonstrated that, fortunately, introducing a single transmission into the RTM would be sufficient to take into account the calibration ringing of the instrument to the nine instrument pixels. Nonetheless, in spectro-imagers such as MTG-S IRS in which a single acquisition consists of 25,600 pixels, the etalon properties could strongly depend on the field of view. Therefore, either the calibration ringing should be removed by a computationally heavy processing using distinct SRFs for every channel and for every pixel or a specific RTM should be used for every pixel, which proves to be impractical for data users.

Consequently, in preparation for the on-ground processing of the MTG-S IRS data and expecting a possible strong impact of calibration ringing specifically on the LWIR products, EUMETSAT developed a mitigation algorithm for the IRS LWIR band. Nonetheless, the methodology can be extended to its MWIR band and to other hyperspectral instruments.

## 3. RTF Uniformisation

As discussed in the introduction, the main contributor to calibration ringing for the CrIS instrument is the band cut-off close to the lower user band limit [6]. For MTG-S IRS, the main contributor is expected to be the etalon, the band cut-off effect being present but not dominating [4]. Therefore, this paper focuses primarily on RTF modulation induced by the etalon effect. We first show with a basic example that additional spectral information is required to be able to correct the calibration ringing in the presence of RTF modulation. Then, we discuss a statistical method to estimate the (non-measured) high frequencies of a spectrum. Finally, we devise a correction factor to be applied to the calibrated measurement to mitigate the calibration ringing.

The presented methodology is referred to as RTF uniformisation as it mitigates the impact of the RTF for all spectral channels as if the RTF was spectrally flat.

### 3.1. First Insight

To gain insight on the impact of calibration ringing and its possible mitigation, we first consider a simple case for which the instrument transmission is modulated at a frequency $f$ and relative amplitude $\alpha$:

$$T(v) = 1 + \alpha \cos[2\pi v f] \tag{3}$$

The interferogram associated with the raw measurement in the presence of RTF modulation $I(x)$ and that associated with the measurement without RTF modulation $I_0(x)$ are given by the Fourier transform of the spectra $Sp(v)$. They are representatives of the frequency decomposition of the spectra:

$$\begin{aligned} I(x) &= FT[Sp.T](x) \\ I_0(x) &= FT[Sp](x) \end{aligned} \tag{4}$$

Inserting the modulated transmission (Equation (3)), we get:

$$I(x) = I_0(x) + \frac{\alpha}{2} \times [I_0(x+f) + I_0(x-f)] \tag{5}$$

Thus, the interferogram recorded in presence of an RTF modulation is given by the superimposition of the one without modulation with two additional low-amplitude interferograms shifted by the modulation frequency, which are called ghosts (Figure 1).

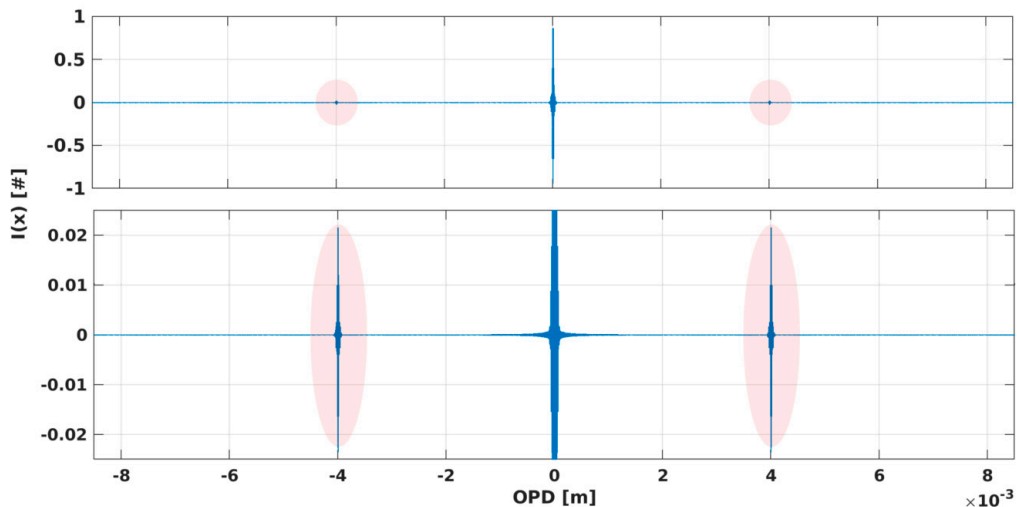

**Figure 1.** Example of simulated interferogram in normalized units of a blackbody in presence of RTF modulation with a periodicity of approximately 250 m$^{-1}$ (modulation frequency of 4 mm) and 5% relative amplitude. The bottom panel is identical to the upper one with the $y$-axis scaled to enhance the ghosts induced by the RTF modulation (underlined by the red ellipses).

Equation (5) can theoretically be reversed to retrieve $I_0(x)$ and thus the spectrum without ringing. At first order in $\alpha$, the approximate correction would be written as:

$$[Sp \otimes SRF](v) \cong FT^{-1}\left[\left\{I(x) - \frac{\alpha}{2} \times [I(x+f) + I(x-f)]\right\} \times Apod(x)\right] \qquad (6)$$

Nonetheless, the samples above the maximum OPD ($OPD_m$) are, in practice, not recorded. As a result, information required to compute the shifted interferograms and to retrieve the corrected spectra from Equation (6) is missing. To achieve this, we would require accessing the interferograms at least until the maximal OPD plus the modulation frequency ($OPD_m + f$).

This simple example shows that any correction should rely on retrieved signal frequencies above the instrument cut-off one, up to $OPD_m + f$ frequencies in this case. In other words, disentangling the RTF variations from the measured spectrum requires bringing higher-frequency information about the input scene. A methodology to achieve this is proposed in the next section. Hereafter, we consider the general problem of any kind of RTF, i.e., not restricted to RTF modulations.

### 3.2. High-Resolution Estimate

The first step of the calibration ringing correction consists thus in estimating, for any calibrated measurement, $Sp_r(v)$, a high-resolution spectrum that is statistically representative of the input scene, $Sp(v)$. This is achieved using the principal components (PCs) decomposition of a dataset of high-resolution spectra $Sp_{ref}(v)$ spanning the range of the viewing angles and of the scene diversity representative of the instrument to correct. This dataset can be generated using an RTM or can consist of measurements from other instruments with higher spectral resolutions.

The principal components are defined as the eigenvectors associated with the greatest eigenvalues of the spectra covariance matrix [11]. The first PC represents the main directions of the signals, called the observation subspace, while the last ones only carry the noise [12]. Ten to a few hundreds of PCs, noted as $N_{PC}$, are generally sufficient to form an orthonormal basis of the high-resolution observation subspace. Noting $PC_{high,n}(v)$ as the nth PC, it is then possible to compute the associated principal components at the instrument resolution and sampling $PC_{low,n}(v)$ by convoluting $PC_{high,n}(v)$ with the instrument SRF and applying a spline interpolation onto the initial measurement sampling grid:

$$PC_{low,n}(v) = SPLINE\{[PC_{high,n} \otimes SRF](v)\} \tag{7}$$

Since the $PC_{low}$ set is generally not an orthonormal basis, we re-normalize the $PC_{high}$ using the inverse of the $PC_{low}$ normalization matrix, noted $N$:

$$N(n,n') = \sum_v PC_{low,n}(v) \times PC_{low,n'}(v), \quad \{n,n'\} \in [1 \dots N_{PC}]^2$$

$$\widetilde{PC}_{high,n}(v) = \sum_{n'=1}^{N_{PC}} N^{-1}(n,n') \times PC_{high,n'}(v) \tag{8}$$

Every measurement is projected onto the basis $PC_{low}$ by computing the associated PC scores (PCS). The high-resolution statistical estimate $Sp_{guess}(v)$ can then be constructed using the same PCS but associated to the high-resolution basis $\widetilde{PC}_{high}$:

$$PCS(n) = \sum_v Sp_r(v) \times PC_{low,n}(v)$$

$$Sp_{guess}(v) = \sum_{n=1}^{N_{PC}} PCS(n) \times \widetilde{PC}_{high,n}(v) \tag{9}$$

As a result, it is possible to estimate high-resolution spectra from lower-resolution measurements bringing statistically relevant high-frequency information. Nonetheless, the method is intrinsically limited; it is of course impossible to perfectly guess information that is not recorded by the instrument.

### 3.3. Correction Factor

Assuming that the estimate $Sp_{guess}(v)$ is an adequate estimation of the input scene $Sp(v)$ and introducing a high-resolution reference of the instrument RTF $T_{ref}(v)$ representative of the actual RTF $T(v)$, we form the correction factor noted as $\gamma(v)$ derived from Equation (2). It aims at cancelling the calibration and at retrieving a corrected spectrum $Sp_c(v)$ close to the input spectrum convoluted with the SRF.

$$\gamma(v) = \frac{[T_{ref} \otimes SRF](v) \times SPLINE\{[Sp_{guess} \otimes SRF](v)\}}{SPLINE\{[Sp_{guess}.T_{ref} \otimes SRF](v)\}}$$

$$Sp_c(v) = Sp_r(v) \times \gamma(v) = \frac{[Sp.T \otimes SRF](v)}{[T \otimes SRF](v)} \times \gamma(v) \cong [Sp \otimes SRF](v) \tag{10}$$

The reference RTF $T_{ref}$ is not expected to vary rapidly and significantly in time. Moreover, it can be seen from Equation (10) that the correction factor is insensitive to low-frequency RTF spectral variations in time, such as those which the presence of ice on the detectors would produce, for example, since these variations would cancel out. Thus, the reference transmission can be considered as quasi-static and would require only occasional updates. It can either be computed by oversampling an average of many calibration slopes if the frequency of its spectral variations is lower than the maximum OPD of the instrument (which is the case for MTG-IRS, for example), or it can otherwise come from instrument modelling or dedicated additional on-ground measurements.

In order to lighten the operational processing, the convolutions of the $\widetilde{PC}_{high}$ and $\widetilde{PC}_{high,n}.T_{ref}$ with the SRF can be pre-computed so that only the PCS for each measurement needs to be calculated. The correction can then be written as:

$$\gamma(v) = \frac{\sum_{n=1}^{N_{PC}} PCS(n) \times V_n(v)}{\sum_{n=1}^{N_{PC}} PCS(n) \times W_n(v)}$$

$$V_n(v) = [T_{ref} \otimes SRF](v) \times SPLINE\{[\widetilde{PC}_{high,n} \otimes SRF](v)\}$$

$$W_n(v) = SPLINE\{[\widetilde{PC}_{high,n}.T_{ref} \otimes SRF](v)\} \tag{11}$$

With this approach, high-resolution guesses of the measurements can be estimated and used to construct a correction factor to mitigate the impact of the calibration ringing.

It is fast enough to be used operationally, as the computational burden imposed by the convolutions can be lightened using pre-computed coefficients.

## 4. Simulations in the Context of MTG-S IRS

In the following, we apply the RTF uniformisation to the case of the MTG-S IRS LWIR band in order to demonstrate the ability and the efficiency of the method in mitigating the calibration ringing.

### 4.1. Simulation Setup

The parameters of the simulation were set such that the simulated instrument resembled the MTG-S IRS LWIR band [3]. Nonetheless, the RTF used for the simulation was made-up specifically for this study and was not expected to be representative of the actual instrument. As shown in Figure 2, it consisted of a smooth door multiplied by a modulation with a periodicity of 250 m⁻¹ (modulation frequency of 4 mm) and a relative amplitude of 5% that was applied to all pixels. As shown in Figure 2, the instrument SRF was plotted along with the RTF, clearly showing that the latter varied at the scale of the SRF, meaning that calibration ringing was expected.

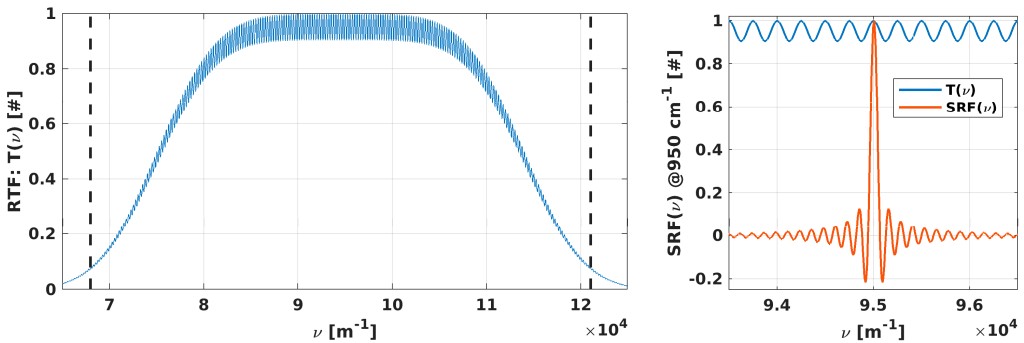

**Figure 2.** (**Left panel**) Radiometric transfer function (RTF) in normalized units as function of the wavenumber of the simulated instrument consisting of a smooth door multiplied by a modulation with a periodicity of 250 m⁻¹ (modulation frequency of 4 mm) and 5% relative amplitude. The dashed lines represent the MTG-S IRS wavenumber range accessible to the users. (**Right panel**) Spectral response function (SRF), including the IRS "light" apodisation centered at 950 cm⁻¹ compared to the RTF.

The dataset to be corrected consisted of a full Earth disc representative of what would be measured from the geostationary orbit at zero degrees longitude by IRS. The Earth disc was covered with 280 step and stare dwells of 40 × 40 pixels, each pixel generating a spectrum. The number of pixels per dwell was downscaled by a factor of 16 compared to IRS to ease the simulations. The input spectra were computed using a radiative transfer model (RTTOV [13]) set up to generate IASI-like spectra between 650 and 1250 cm⁻¹ with a sampling of 0.25 cm⁻¹ for 14 July 2022 at 12:00 UTC using ECMWF forecasts as inputs. First, the IASI numerical apodisation was removed at the interferogram level. Then, the simulated IASI spectra were multiplied by the RTF, and applying a Fourier transform allowed for computing the associated raw interferograms. Computing the MTG-S IRS-like spectra was then a matter of applying the IRS numerical "light" apodisation and performing an inverse Fourier transform. Calibration views were generated along with the Earth views and were used for the radiometric calibration of the simulated products in a processing chain similar to the one that will be routinely operated [3]. Figure 3 shows the LWIR average spectral radiance distribution over the full disc and a selection of the sampled calibrated spectra, as in reality, at 0.61 cm⁻¹. It is worth noting that the simulation was run without radiometric noise to emphasize the impact of the sole calibration ringing.

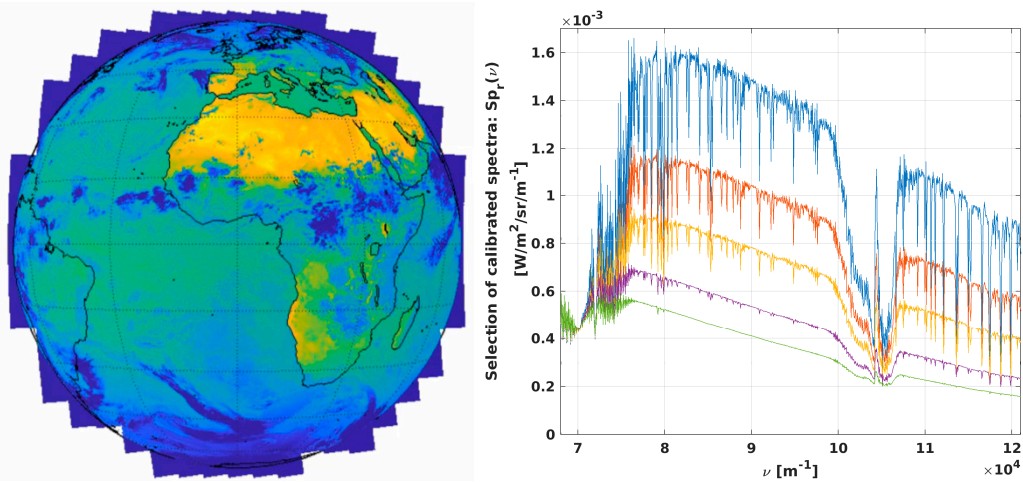

**Figure 3.** (**Left panel**) LWIR broad-band radiance distribution of a full scan of MTG-S IRS consisting of 280 step and stare dwells. (**Right panel**) Selection of calibrated spectra over MTG-S IRS LWIR band representative of the simulation diversity.

### 4.2. RTF Uniformisation Dataset

The RTF uniformisation dataset was composed of actual IASI measurements acquired from the Metop-C platform [1]. One full day of measurements on 1 January 2023 was used, yielding approximately 1,200,000 spectra. The IASI measurements were supersampled enough with respect to IRS, as the IASI maximum OPD equals 20 mm, which is significantly larger than the sum of the IRS maximum OPD (8.2 mm) and the transmission modulation frequency (4 mm), which is the criteria for an efficient correction, as discussed in Section 3.1. The IASI spectra were then cropped to match the IRS LWIR band, and the IASI numerical apodisation was removed.

As described in Section 3.2, we computed the spectra covariance matrix and derived the values of $PC_{high}$ and associated $PC_{low}$. The high-resolution spectra covariance and the first three $PC_{high}$ and $PC_{low}$ values are represented in Figure 4.

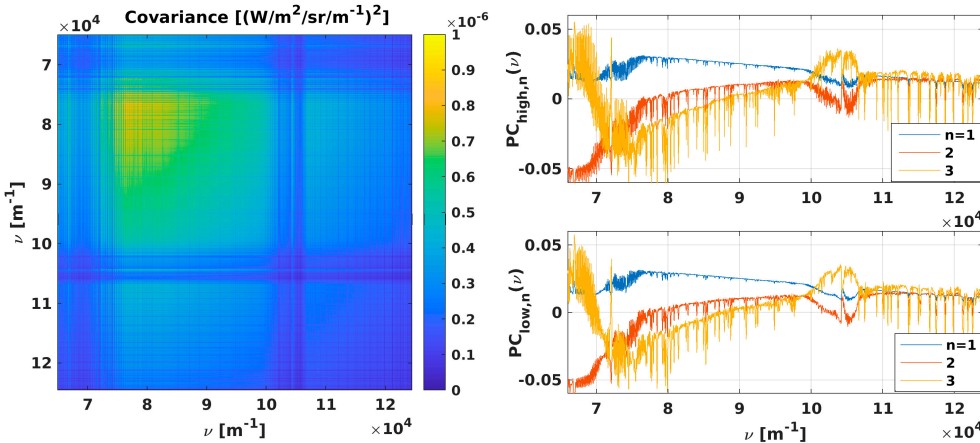

**Figure 4.** (**Left panel**) High-resolution covariance matrix over MTG-S IRS LWIR band. (**Right panel**) First three high- and low-resolution principal components (PCs).

### 4.3. RTF Uniformisation Efficiency

Choosing to construct the high-resolution dataset from actual IASI measurements and not from the same reference spectra used in the IRS simulation of Section 4.1 avoided having perfect estimates by design and aimed at proving that the efficiency of the RTF uniformisation was not overly sensitive to the choice of high-resolution dataset. Two simulations with and without the RTF uniformisation were run over the full IRS scan. The

resulting calibrated spectra were then compared by subtracting them to a common reference free of calibration ringing. The derived radiometric errors were converted into equivalent temperature errors dividing by the derivative of the Planck function at the temperature of reference of 280 K.

The minimum, maximum, and average calibration ringing error as a function of the wavenumber are presented on Figure 5. This highlighted the strong scene dependency of the calibration ringing. The calibration ringing was characterized by error spikes up to ±1 K. After the RTF uniformisation using 10 PC, a perfect cancellation of the average radiometric error and a strong reduction in the variability was observed.

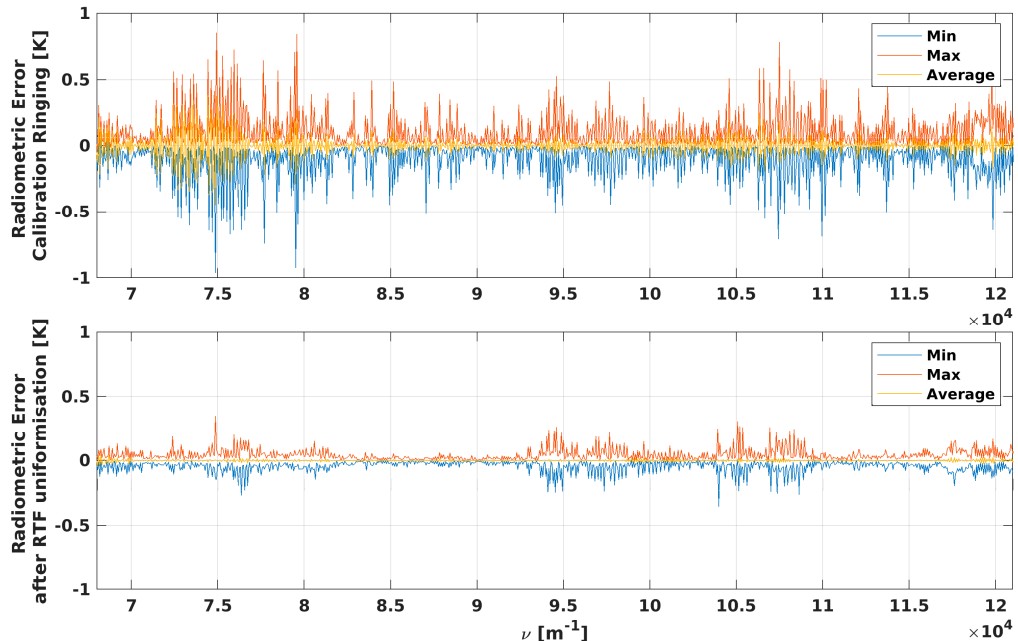

**Figure 5.** (**Upper panel**) Minimum, maximum, and average radiometric error in equivalent temperature error in Kelvin due to the calibration ringing. (**Bottom panel**) Same metrics after application of the RTF uniformisation using 10 principal components (PCs).

Figure 6 shows the error standard deviation over the band computed from all the spectra and expressed as equivalent temperature error as a function of the number of PCs. The calibration ringing produced errors of 100 mK standard deviation. As soon as the RTF uniformisation was activated, this value rapidly decreased, even if only a couple of PCs were used, and it then reached a plateau for which the error standard deviation was divided by a factor 10. It is worth noting that the error increased again when the number of PCs reached 50.

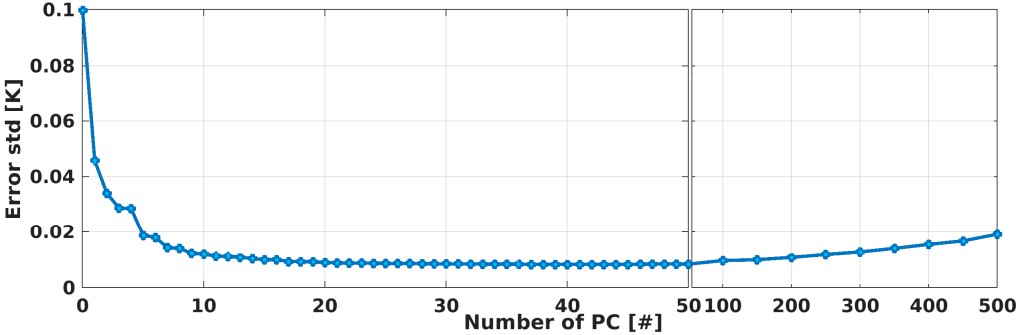

**Figure 6.** Residual ringing error standard deviation in Kelvin as function of the number of principal components (PCs) used by the RTF uniformisation.

As discussed in Section 3.2, the efficiency of the correction was naturally hampered by the intrinsic limitation in estimating the high-resolution spectrum. Nonetheless, the RTF uniformisation appeared to be particularly efficient and well suited for an operational mitigation of the calibration ringing that could be observed in the MTG-S IRS LWIR band.

## 5. Discussion

There are still several points of limitations of this study and remarks for the future of calibration ringing mitigation in Fourier transform spectrometers:

- As expected for MTG-S IRS, the RTF may vary between the pixels; therefore, the correction and the required pre-computations of Section 3.3 would become pixel-based.
- In view of the operational implementation of the RTF uniformisation by EUMETSAT for MTG-S IRS, the choice of the high-resolution dataset to use is open. The best candidates are Metop IASI [1], as presented in this study, which is considered as an international reference, or its successor Metop-SG IASI-NG [14] (foreseen in 2025). The main limitation of these instruments is that their maximum sounding angles (up to 50°) are smaller than IRS (up to 90° close to Earth's rim); therefore, the dataset could be complemented with RTM simulations at high-sounding angles.
- The RTF uniformisation is based on the current knowledge of instrument transmission. Thus, a careful monitoring of the calibration slopes evolution in time is needed as well as updating the parameters of the algorithm if required. This technique would fail if, for example, the etalon characteristics were to rapidly fluctuate in time; nonetheless, this is not expected for MTG-S IRS.
- In real conditions, measurements are noisy; therefore, the high-resolution estimate can be adapted by introducing a radiometric noise normalization into the PC projection of Section 3.2. This point was not discussed in this study. Nonetheless, the number of PCs to use and the RTF uniformisation efficiency are not expected to be strongly impacted.
- The RTF uniformisation methodology is not specific to IRS LWIR band; it can be extended to its MWIR band and other hyperspectral instruments. It is also expected to be efficient in mitigating the calibration ringing induced by band cut-off, as for the CrIS instrument [6].
- The high-resolution statistical estimate approach introduced in Section 3.2 is actually applicable to other hyperspectral instruments. It would help in creating statistically relevant high-resolution datasets to test algorithms of a new generation of satellites before launch.

## 6. Conclusions

In this paper, the fundamentals of calibration ringing error were recalled: a radiometric error propagated to the calibrated radiances acquired by FTS caused by significant spectral variations in the instrument's RTF at the scale of the instrument SRF. A mitigation strategy, called RTF uniformisation, relying on a high-spectral-resolution estimate of the measurements using principal components decomposition, was introduced. Finally, the RTF uniformisation efficiency was assessed in the context of the MTG-S IRS LWIR band.

The RTF uniformisation appeared to be sufficiently efficient and computationally inexpensive to be implemented into operational processing. It will be activated on day one for MTG-S IRS as an additional processing applied to the LWIR-calibrated radiances. In doing so, a reduction by a factor 10 in the radiometric errors induced by calibration ringing is expected.

**Author Contributions:** Conceptualization, B.T. (Bernard Tournier), C.S. and P.D.; investigation and writing, P.D.; coding, P.D. and G.D.; review and editing, G.D., D.C., and B.T. (Bertrand Theodore); project administration, D.C. All authors have read and agreed to the published version of the manuscript.

**Funding:** This research was funded by EUMETSAT.

**Data Availability Statement:** The data used in this study are described in Section 4 and accessible on the EUMETSAT data center (https://www.eumetsat.int/eumetsat-data-centre, accessed on 30 May 2023).

**Acknowledgments:** We would like to thank Dave Tobin (CIMSS) and Nigel Atkinson (Met Office) for the initial discussion on this topic.

**Conflicts of Interest:** The authors declare no conflict of interest.

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
