# Peer review of "Mitigation of Calibration Ringing in the Context of the MTG-S IRS Instrument"

_remotesensing, doi:10.3390/rs15112873_

Round 1

Reviewer 1 Report

See attached review

Author Response

We thank the reviewer for his/her time and positive feedbacks. We have updated the manuscript correcting all typos and accounting for his/her comments:

  • We have introduced the “InfraRed Sounders” denomination in the abstract and section 1.

  • We have updated the SRF definition paragraph of section 2 as such: “The instrument SRF, noted SRF(nu) as function of the wavenumber nu, is given in the context of Fourier transform spectrometry by the Fourier transform of the numerical apodisation Apod(x) applied to the recorded interferograms as function of the Optical Path Distance x (OPD) times potential wavenumber dependent self-apodisations induced by instrumental defects. We assume hereafter for simplicity that the self-apodisation defects are negligible or perfectly compensated by dedicated software corrections such that the SRF is the same for all spectral channels.”

  • The “4mm frequency” is replaced by “periodicity of 250 m-1 (modulation frequency of 4 mm)” in the text of section 4.1 and in the label of figure 2.

  • The left panel of figure 3 is indeed equivalent to the LWIR broad-band radiance image. We have updated the label to: “LWIR broad-band radiance distribution”

  • Yes, the RTF uniformisation will be implemented for day-1, we have updated the conclusion as such: “The RTF uniformisation appears sufficiently efficient and computationally inexpensive to be implemented into an operational processing. It will be activated on day-1 for MTG-S IRS as an additional processing applied to the LWIR calibrated radiances.”

In attached file, the version including all reviewer comments.

Thank you, regards.

Reviewer 2 Report

1.    Line 34:What is the instrumental radiative transfer function that needs to be defined or cited? Is there a similar statement in other reference paper.

2.    Line 71:What is the cause of self-apodisation and what defects does it bring? These need to be explained.

3.    Line 77:Is the radiometric calibration factor the same as the radiometric calibraion coefficient slope? If there are generic descriptions and definitions, please try to keep them consistent with those in the internationally published literature.

Line 108:There is no text in the full text to explain what is RTF uniformisation, why is there exist non-uniformity, and how big is the error caused by non-uniformity.

4.      Line 111:”For MTG-SIRS,the main contributor is expected to be the etalon, the band cut-off effect beingpresent but not dominating”What is the basis of this conclusion?

5.    Line 113:Since this paper focuses on RTF modulation brought about by the etalon effect, it is necessary to explain clearly what is the etalon effect first, and how is it produced?

6.    Line 213:Is the modulation different for different detectors? Is the modulation more different between targets or between detectors? This conclusion needs to be verified.

Author Response

We thank the reviewer for his/her time and suggestions. We have updated the manuscript accounting for his/her comments:

  • 1- The Radiometric Transfer Function (RTF) is the overall transmission of the instrument as function of the wavenumber; the references [5] and [6] use the term spectral instrument responsivity instead. Thus, we propose to add a statement to make the link between the two denominations in section 1: “Such effect arises when the instrument radiometric transfer function (RTF), sometimes also referred to as spectral instrument responsivity, varies significantly within the domain of the instrument spectral response function (SRF).”

  • 2- We have updated the second paragraph of section 2 and added 2 references to examples of self-apodisation effects and their corrections in the context of CrIS and IASI: “The instrument SRF, noted SRF(n) as function of the wavenumber n, is given in the context of Fourier transform spectrometry by the Fourier transform of the numerical apodisation Apod(x) applied to the recorded interferograms as function of the Optical Path Distance x (OPD) times potential wavenumber dependent self-apodisations induced by instrumental defects. We assume hereafter for simplicity that the self-apodisation defects are negligible or perfectly compensated by dedicated software corrections such that the SRF is the same for all spectral channels [9,10].”

  • 3- Yes they are the same. We recognise that the “Radiometric calibration factor” denomination is ambiguous and accept to update with “radiometric calibration slope coefficient” instead, in section 2.

  • The “RTF uniformisation” denomination is described at the end of section 3: “The methodology is referred to as RTF uniformisation as it mitigates the impact of the RTF for all spectral channels, as if the RTF was spectrally flat.“ We have decided to move the sentence early in the manuscript, at the beginning of section 3.
    Examples of errors caused by the non-uniformity of the RTF are presented in section 4.

  • 4- Indeed, this statement is not backed-up as the exact knowledge of the transmission cannot be disclosed. Nonetheless, the reference 4 hints toward a much more sensitivity to transmission modulations than gradients.

  • 5- We define the etalon in the section 1 with a link to the Wikipedia page: “As a second example, RTF modulations can arise from unexpected light loop between optical surfaces. Optical elements in transmission such as lenses, windows or protective layers with non-perfect coating can create low finesse etalons [7]; as a result the transmission appears modulated as function of the incident light wavenumber. The latter effect is expected in the MTG-S IRS instrument, with a spectral modulation with a periodicity of approximately 250 m-1 (modulation frequency of 4 mm) and up to a few percent relative amplitude. The optimization of the optical design of the instru-ment (e.g. re-designing the optical coating such that the RTF is flat) proved technically out of reach. “

  • 6- We propose to add a bullet in the discussions to mention the impact of having different RTF for each pixel: “As expected for MTG-S IRS, the RTF may vary between the pixels, therefore the correction and the required pre-computations of section 3.3 are pixel-based.”

In attached file, the manuscript including all reviewers comments,

 Thank you, regards.

Reviewer 3 Report

This is a very good paper.  It is covering an important topic, is technically excellent, and is presented in a very clear and organized manner.  I have four minor comments/questions:

1. Parts of the introduction include: “These variations generate distortions of the SRF [4], which, if unaccounted, propagate as radiometric errors into the calibrated Earth 36 view spectra exploited by the users.” And “RTF spectral variations are actually expected for most instruments.” And “Calibration ringing errors occur when the radiometric calibration fails to perfectly compensate for the RTF spectral variations.”  

Implicit in these statements is that the final calibrated spectra for interferometers like IASI, CrIS, and IRS are provided on a standard wavenumber scale and SRF.  In this context if the processing is not able to perform the accurate processing to transform the original data with its raw SRF to the calibrated grid and SRF, then this can be considered an error.  But this approach is not always typical, or even attempted.  (If the data was not processed to be on a standard grid with standard SRF, and provided to users on its original scales, it might be as useful to users, but it would not contain associated errors).  The point that I think is worth making as part of the introduction is that is really only for interferometers that this type of approach is even feasible, as it is not feasible to transform grating spectrometer (e.g. AIRS) or broadband filter spectrometers (e.g. VIIRS) to even attempt to produce data with common SRFs across pixels or sensors.  I think some type of additional intro material about the process of producing data with a common grid and SRF would help the general reader, and put this important work into context.

2.  It is not clear or explained why a spline (or any other) interpolation is needed in Equation 7.  Can’t the lower instrument resolution PCs be obtained by convolving the high resolution PCs with the instrument SRF?  Please add a sentence or two to explain why the interpolation is needed.  

3.  Section 4.  Please indicate if the user SRF is unapodized, or is the sinc with light apodization.  If lightly apodized, it might be good to indicate the size of residual errors that are found after correction for unapodized.

4.  Section 4.3.  For the choice of IASI data for the high resolution training, this includes a limited set of view angles compared to IRS.  Is there a consideration needed for the larger IRS view angles and is this covered in the anticipated residual errors shown in Figure 5?  Do the errors on the bottom panel of Figure 5 correspond to any physical parameters such as the larger IRS view angles?

Author Response

We thank the reviewer for his/her time, positive feedbacks and suggestions. We have updated the manuscript accounting for his/her comments and we provide complementary information below:

  1. It is indeed a very good point, therefore, we propose two updates in section 2:

“Thus, if the optical transmission varies significantly at the scale of the SRF then the calibrated spectrum does not equal the input spectrum convoluted with a unique, pixel-independent SRF, as desired by the users of FTS based multi-detector data products like those of IASI, CrIS, and IRS.”

“As discussed in [6], the effect of calibration ringing in data applications does not necessarily constitute an error if SRF distortion (cf. Equation (2)) is explicitly accounted for in the radiative transfer models (RTM). This is even without alternative for grating spectrometers (like AIRS), for which the generation of data to a common spectral response across detectors is unfeasible. However, if the transmission varies between detectors, the SRF would become detector-dependent. In the case of the CrIS instrument, Borg et al [6] have demonstrated that, fortunately, introducing a single transmission into the RTM would be sufficient to take into account the calibration ringing of the instrument to the nine instrument pixels. Nonetheless, in spectro-imagers such as MTG-S IRS in which a single acquisition consists of 25600 pixels, the etalon properties could strongly depend on the field of view. Therefore, either the calibration ringing should be removed by a computationally heavy processing using distinct SRFs for every channel and for every pixel, or a specific RTM should be used for every pixel, which proves unpractical for the data users.”

  • The spline just allows re-sampling the convolution at high-resolution sampling to the measurement grid. We have updated the sentence to: “Noting PC_(high,n) (ν) the nth PC , it is then possible to compute the associated principal components at the instrument resolution and sampling PC_(low,n) (ν) by convoluting PC_(high,n) (ν) with the instrument SRF and applying a spline interpolation onto the initial measurement sampling grid.”

  • We have added the mention to the light apodisation in section 4.1 and figure 2 caption. We did not run the simulation without apodisation; nonetheless the difference is expected to be quite small as studied in reference [4].

  • 4. This issue is acknowledged in section 5, in the presented results, part of the residual errors could indeed arise from the “high-resolution” IASI dataset limitation not covering high-sounding angles (above 55 degrees). This hypothesis has not been evaluated yet, but we intend to do it in later testing.

In attached file, the updated manuscript including all reviewers comments,

Thank you, regards.

Reviewer 4 Report

Please, see attached file for comments & suggestions.

Author Response

We thank the reviewer for his/her time and suggestions. We have updated the manuscript accounting for his comments and we provide complementary information below:

  • Title: “Mitigation of Calibration Ringing in the Context of the MTG-S IRS Instrument”

  • As discussed in section 5: “The RTF uniformisation is based on the current knowledge of the instrument transmission. Thus, a careful monitoring of the calibration factors evolution in time is needed as well as updating the parameters of the algorithm if required. This technique would fail if for example the etalon characteristics would rapidly fluctuate in time, nonetheless that is not expected for MTG-S IRS.”

From the available information, the etalon is not expected to vary in time and that will be confirmed soon during vacuum tests. We are currently preparing a daily monitoring of its properties and an automatic derivation of the expected radiometric errors. In case of strong variations, we will be able to update the correction parametrisation at level of the month, which is expected to be well enough.

Concerning the availability of the IASI data, there is no constraint as the same training dataset can be used for several years. We will only update the PC basis if a better dataset, for example IASI-NG, becomes available.

Therefore, we expect no discontinuity of the RTF uniformisation correction and no impact to the end-users.

  • As discussed in section 4.2, the dataset has to be diverse, we use 1200000 IASI spectra covering a full day of measurements and therefore covering a large diversity of atmospheric scenes. In the future, as discussed in section 5, we aim at using IASI-NG spectra complemented with Radiative Transfer model simulations for higher sounding angles not covered by both IASI and IASI-NG, to even increase the representativity of the training set.

  • We have tested successfully the methodology in noisy situations, nonetheless we did not want to detail this case here to lighten the paper. We propose to add a comment about the number of PC required in noisy situations in section 5: “In real conditions, measurements are noisy; therefore, the high-resolution estimate can be adapted introducing a radiometric noise normalization into the PC projection of section 3.2. This point is not discussed in this study. Nonetheless, the number of PC to use and the RTF uniformisation efficiency are not expected to be strongly impacted.”

  • The use of different colours in Fig. 3 (right panel) has no other purpose than to better represent some individual spectra within the typical dynamic range. We wanted to show what typical IRS LWIR measurements would look like for readers unfamiliar with the IRS/LWIR instrument sampling and resolution.

In attached file, the updated manuscript including all reviewers comments,

Thank you, regards.
